# Islamic and capitalist economies: Comparison using econophysics models of wealth exchange and redistribution

**Takeshi Kato** [ORCID] *

Hitachi Kyoto University Laboratory, Open Innovation Institute, Kyoto University, Kyoto, Japan

* kato.takeshi.3u@kyoto-u.ac.jp

**Data Availability Statement:** All relevant data are within the paper.

**Funding:** The author received no specific funding for this work.

## Abstract

Islamic and capitalist economies have several differences, the most fundamental being that the Islamic economy is characterized by the prohibition of interest (*riba*) and speculation (*gharar*) and the enforcement of *Shariah*-compliant profit–loss sharing (*mudaraba*, *murabaha*, *salam*, etc.) and wealth redistribution (*waqf*, *sadaqah*, and *zakat*). In this study, I apply new econophysics models of wealth exchange and redistribution to quantitatively compare these characteristics to those of capitalism and evaluate wealth distribution and disparity using a simulation. Specifically, regarding exchange, I propose a loan interest model representing finance capitalism and *riba* and a joint venture model representing shareholder capitalism and *mudaraba* of an Islamic profit–loss sharing system; regarding redistribution, I create a transfer model representing inheritance tax and *waqf* of an Islamic wealth redistribution system. As exchanges are repeated from an initial uniform distribution of wealth, wealth distribution approaches a power-law distribution more quickly for the loan interest than the joint venture model; and the Gini index, representing disparity, rapidly increases. The joint venture model's Gini index increases more slowly, but eventually, the wealth distribution in both models becomes a delta distribution, and the Gini index gradually approaches 1. Next, when both models are combined with the transfer model to redistribute wealth in every given period, the loan interest model has a larger Gini index than the joint venture model, but both converge to a Gini index of less than 1. These results quantitatively reveal that in the Islamic economy, disparity is restrained by prohibiting *riba* and promoting reciprocal exchange in *mudaraba* and redistribution through *waqf*. Comparing Islamic and capitalist economies provides insights into the benefits of economically embracing the ethical practice of mutual aid and suggests guidelines for an alternative to capitalism.

## Introduction

The unequal distribution of wealth has created disparities in income and assets and has become a major social issue worldwide. The Gini index, which represents disparity, is gradually rising in the major developed countries in the Organisation for Economic Co-operation and Development (OECD) [1]. A Gini index of 0.4 is considered a warning level for social unrest [2], but there are many countries in the world where the coefficient exceeds 0.4 [3]. The

**Competing interests:** The author have declared that no competing interests exist.

United Nations Sustainable Development Goals are also concerned with disparity and its effects, specifically Goal 10, which aims at reducing inequality, and also Goal 1 (no poverty), Goal 2 (zero hunger), Goal 3 (good well-being), Goal 8 (inclusive economic policy), and Goal 16 (justice) [4].

According to David Graeber, an anthropologist and activist, the past 5,000 years of human history have alternated between cycles of a bullion-based monetary economy and a virtual money-based credit economy [5]. The monetary economy is characterized by interest-bearing debt, war, and slavery, while the credit economy tends to be associated with a peaceful, moral society. These various economic cycles include the age of agriculture (credit economy), the Axial Age (monetary economy), the Middle Ages (credit economy), and the age of the great capitalist empires (monetary economy), culminating in the modern era with the transition from the gold standard to a floating currency system. Today, society is transitioning from a monetary economy to a credit economy, but as debt and war are still widespread, it is impossible to move on to the next alternative to capitalism.

During the Middle Ages, an era of a credit economy, moral and financial innovations emerged from the Islamic world [5]. The Islamic code (*Shariah*) outlawed interest-bearing loans that made the proliferation of money a self-purpose and encouraged bankers and merchants to engage in credit operations. Joint management, in which investors and operators shared profits and losses with each other, was favored in finance as an extension of mutual aid, and labor arrangements were based on profit-sharing. These practices developed into an economic framework for Islamic countries. Thus, the Islamic economy is characterized by the prohibition of *riba* (interest) and *gharar* (speculation) as an economy based on real transactions, the enforcement of *mudaraba* (joint venture) and *murabaha* (profit-sharing through agreed-upon contracts) as a face-to-face economy, and the promotion of *zakat* (charity) and *waqf* (donation) as an economy embedded into religion in the *ummah* (community) [6–8]. The Islamic economy can be considered a reference for transforming the modern world from capitalism to the next credit economy and creating an equal and free society with reduced disparity.

In econophysics, a new research field in which economic phenomena are approached from the perspective of physics, the distribution of wealth has been investigated using multi-agent exchange models based on the exchange of kinetic energy between two ideal gas particles [9–11]. In these exchange models, various wealth distributions emerge, including exponential, power-law, and delta distributions, depending on parameters such as the amount of exchange between the two agents, the savings rate, and the stock contribution rate. Econophysics exchange models can be used to explain the basic causes of various distributions in real society; for example, empirical laws such as the Pareto principle [12] for the distribution of income and Zipf's law [13] for the distribution of city size, respectively. Therefore, applying an econophysics approach to examine the Islamic economy and capitalism can allow key differences between the two to be determined.

Thus, this study aims to identify the fundamental differences between Islamic and capitalist economies from the perspective of econophysics to gain insight that can help guide capitalism's transition into an alternative credit economy. To do so, I create new econophysics models that represent both the Islamic economy and capitalism and compare the two by simulating wealth distribution and disparity based on these models. The current study is novel in that it incorporates loan interest and profit/loss distribution in joint ventures into mathematical models, which expands on previous models [9–11] that have incorporated only the exchange of wealth. The new models also consider redistribution through transfers, which has been disregarded in previous models. Moreover, no previous studies have compared Islamic and capitalist economies using the framework of econophysics, which reveals the differences between

the two from a physical and quantitative perspective, rather than the traditional qualitative one. The study also provides new insights regarding the next credit economy after the Middle Ages, which will bring equality in society.

The new models I propose incorporate Karl Polanyi's three economic modes (reciprocity, redistribution, and market exchange) [14]; this approach goes beyond traditional models that solely involve the exchange of wealth. With regard to exchange, I focus on loan interest by examining finance capitalism and *riba* (interest), as well as shareholder capitalism and reciprocal joint ventures (*mudaraba* of a *Shariah*-compliant profit–loss sharing system), and I model these as the distribution of profits and losses in exchange. In terms of redistribution, I model a compulsory inheritance tax, which is based on capitalism's centralization of power as shown by Polanyi, and reciprocal donation (*waqf* of a *Shariah*-compliant wealth redistribution system), which is based on noncentralized aid in a community, as transfers in every predetermined period (although *waqf* in actuality is the act of relinquishing ownership of assets to be held in trust, this study considers it a transfer of wealth to the public, i.e., to the community as a whole).

The remainder of this paper is organized as follows. The next section presents a literature review on exchange models in econophysics, and the Methods section presents new exchange and redistribution models. The simulation results of wealth distribution and the calculation of disparity, represented by the Gini index, as well as a comparison of the characteristics of the Islamic economy and capitalism, are presented in the Results section. The Discussion section revisits the contemporary significance of the Islamic economy in light of the results and discusses the credit economy as the next alternative to capitalism. Finally, the last section presents conclusions and future challenges.

## Literature review

This section provides a brief literature review on exchange models in econophysics based on Chakrabarti et al. [10] and Kato et al. [11].

In 1906, economist Vilfredo Pareto determined that the distribution of income follows a power law [12]. This came to be known as the "Pareto principle," which states that income is concentrated in a few wealthy people [15]. Champernowne further explained the Pareto principle in 1953 using a model in which the income distribution changed over time through a stochastic process [16]. Later, based on a review of a number of studies in 2009, Yakovenko and Rosser suggested that the distribution of income and wealth is consistent with a lognormal or gamma distribution and the tail of the distribution follows a power law [9].

In 1986, sociologist John Angle showed that a gamma distribution arises from a stochastic process in which two economic agents contribute to each other's wealth surplus, excluding savings, and one randomly chosen agent receives all the amount contributed [17]. Using a model of kinetic energy exchange in collisions of ideal gas particles, Hayes [18] and Chakraborti [19] showed that a delta distribution arises in which wealth is concentrated in a single economic agent. This concentration of wealth is because the amount of wealth contributed is determined according to the poorer of the two economic agents.

Subsequently, several exchange models were proposed that extended the models of Angle, Hayes, and Chakraborti. For example, in Chakrabarti et al.'s [10] model, an exponential distribution is obtained if both agents divide the contribution by random allocation rather than one agent receiving the entire contribution. Chatterjee et al.'s [20] model results in a gamma distribution by randomly dividing contributions with a constant savings rate for all economic agents. In another model by Chatterjee et al. [21], a power-law distribution is obtained if the savings rate for all economic agents follows a uniform distribution.

To restrain wealth disparity, models that introduce the concepts of taxation and insurance have been proposed. In Guala's [22] taxation model, a fixed tax rate is imposed, and the total tax is distributed equally among all economic agents after a random division of the exchange. As the tax rate increases, the distribution shifts from an exponential distribution to a gamma distribution and then reverts to an exponential distribution. In Chakrabarti et al.'s [23] insurance model, economic agents insure against risk; after an exchange, the winner transfers a portion of the surplus to the loser at a constant rate. As the transfer rate increases, the distribution shifts from an exponential distribution through a gamma distribution to a delta distribution.

In addition to these models, other models that consider regions and surplus stocks have been proposed. The regional model from Kato et al. [24] introduces a spatial exchange range and a regional support bias (providing an advantageous probability for poorer regions) in addition to the regional economic circulation rate (savings rate). The narrower the exchange range and the larger the bias, the closer to a normal distribution. In Kato et al.'s [11] surplus stock model, the wealthy contribute surplus stock in addition to matching the contribution of the poor, which is divided randomly. As the surplus stock contribution rate increases from 0 to 1, the distribution changes from a delta distribution to a gamma distribution (as in Chatterjee et al.'s [20] model). The model also shows that the contribution of surplus stock by the wealthy is necessary to both stimulate the economy and reduce disparity.

Thus far, I have outlined various econophysics exchange models. In all these conventional models, a random exchange of wealth is assumed. With this in mind, the purpose of this study is not to improve upon the conventional models but to construct new models for Islamic and capitalist economies. Therefore, for the first time, I am modeling a) loan interest (borrower's interest and lender's interest, borrower's profit/loss burden) and joint venture (profit/loss allocation between two agents) in wealth exchange and b) transfers (redistribution in every predetermined period) in wealth redistribution rather than random exchange, as in conventional models.

Additionally, regarding the characteristics of the Islamic economy, this study mainly refers to Nagaoka [6,7] and Kato [8], in addition to some recent literature.

Kamdzhalov [25] criticized financial capitalism for not creating real value and stated that new technologies such as blockchain and fintech increase trust in the interest-free and profit-loss sharing framework of Islamic finance. Arfah et al. [26] highlighted the harmful effects of debt and speculation in a capitalist economy and recommended an Islamic economy for the post-COVID-19 period, one that is based on sharing profits and losses and redistribution through *zakat* and *waqf* according to the principles of justice and fairness. Yukti et al. [27] stated that a smooth money flow will lead to a healthy economy and mitigate the global economic crisis after COVID-19, the distribution of wealth should be broadened through *mudaraba* and *waqf*, and the Islamic economy can provide an alternative solution to capitalism.

## Methods

### Exchange model

To begin, I use the basic Chatterjee et al. [20] model as the exchange model. I refer to it here as the random exchange model (R model) for convenience. In the R model, two agents $i, j$ (= 1, 2,···,$N$) are randomly selected from among $N$ economic agents. Let $m_i(t)$ denote the wealth of agent $i$ at time $t$ and $m_j(t)$ denote the wealth of agent $j$. As shown in Fig 1B, two agents, $i$ and $j$, save a portion of their wealth at time $t$ at a common savings rate $\lambda$ and exchange the remaining wealth, $(1-\lambda)\cdot(m_i(t)+m_j(t))$, excluding savings, with random division probability $\varepsilon$. $\varepsilon$ is a uniform random number defined in the range $0 \leq \varepsilon \leq 1$. The respective wealth $m_i(t+1)$ and $m_j(t+1)$

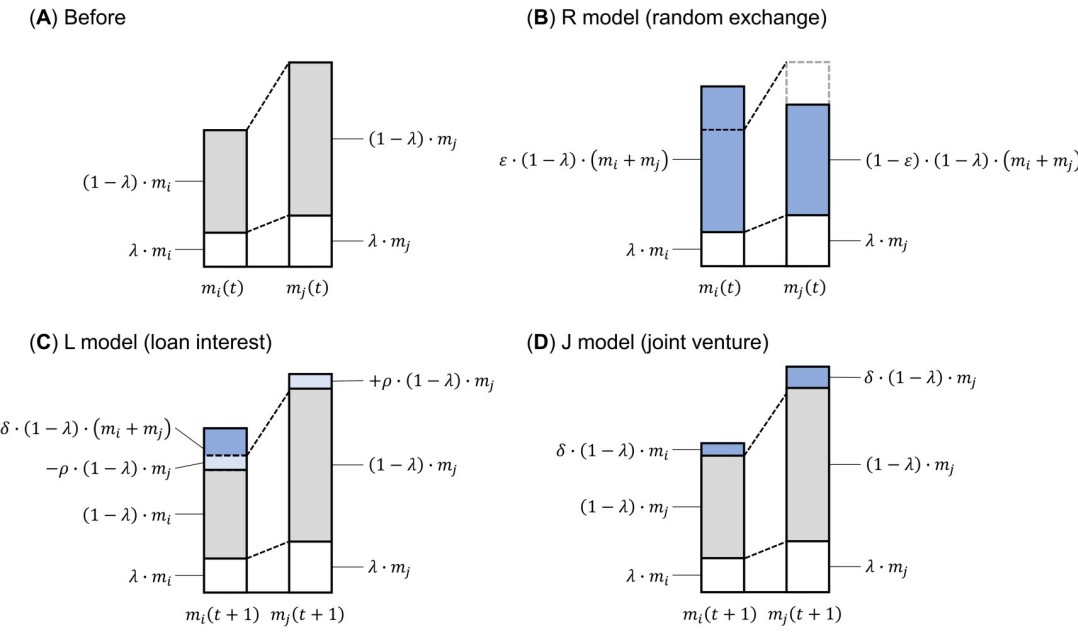

**Fig 1. Exchange models.** (A) Before exchange, (B) R model (random exchange), (C) L model (loan interest), and (D) J model (joint venture). $\lambda$ is savings rate, $\varepsilon$ is division probability, $\rho$ is interest rate, and $\delta$ is profit/loss rate.

of the two agents *i and j* at time *t*+1 is expressed as Eq (1):

$$m_i(t+1) = \lambda \cdot m_i(t) + \varepsilon \cdot (1-\lambda) \cdot (m_i(t) + m_j(t));$$
$$m_j(t+1) = \lambda \cdot m_j(t) + (1-\varepsilon) \cdot (1-\lambda) \cdot (m_i(t) + m_j(t)). \tag{1}$$

Based on the R model, I construct a loan interest model corresponding to financial capitalism. I refer to it here as the L model for convenience. In the L model, the interest rate $\rho$ and the profit/loss rate $\delta$ are newly set in addition to the savings rate $\lambda$. $\delta$ is a uniform random number defined in the range $-\delta_w \leq \delta \leq \delta_w$ ($\delta_w \geq 0$). As shown in Fig 1C, two agents *i and j* are randomly selected from *N*; agent *i* is the borrower, and the other agent *j* is the lender. Borrower *i* pays interest $(1-\lambda) \cdot \rho \cdot m_j(t)$ and bears profit/loss $(1-\lambda) \cdot \delta \cdot (m_i(t) + m_j(t))$, while lender *j* earns interest $(1-\lambda) \cdot \rho \cdot m_j(t)$. The wealth $m_i(t+1)$, $m_j(t+1)$ of the two agents *i and j* at time *t*+1 is expressed as Eq (2):

$$m_i(t+1) = \lambda \cdot m_i(t) + (1-\lambda) \cdot (m_i(t) - \rho \cdot m_j(t) + \delta \cdot (m_i(t) + m_j(t)));$$
$$m_j(t+1) = \lambda \cdot m_j(t) + (1-\lambda) \cdot (1+\rho) \cdot m_j(t). \tag{2}$$

I also create a joint venture model to examine shareholder capitalism and *mudaraba* of the Islamic economy, hereinafter the J model. Compared to the relationship between shareholders and operators, in *mudaraba*, the joint operators are actively involved with each other in a partnership, but both are represented by the same model mathematically. In the J model, the interest rate $\rho$ in the L model is not used, only the profit/loss rate $\delta$ is used. As shown in Fig 1D, two agents *i and j* are randomly selected from *N*; they obtain a profit/loss $(1-\lambda) \cdot \delta \cdot (m_i(t)$ and $(1-\lambda) \cdot \delta \cdot (m_j(t)$, respectively, depending on the profit/loss rate $\delta$. The respective wealth $m_i(t+1)$ and

$m_j(t+1)$ of the two agents $i$ and $j$ at time $t+1$ is expressed as Eq (3):

$$m_i(t + 1) = \lambda \cdot m_i(t) + (1 - \lambda) \cdot (1 + \delta) \cdot m_i(t);$$

$$(3)$$

$$m_j(t + 1) = \lambda \cdot m_j(t) + (1 - \lambda) \cdot (1 + \delta) \cdot m_j(t).$$

## Redistribution model

Next, with respect to redistribution, which is not considered in the traditional exchange model, I construct a transfer model that corresponds to the inheritance tax of capitalism and *waqf* of the Islamic economy, hereinafter the T model. While inheritance taxes are levied based on centralized power in capitalism, *waqf* is self-initiated based on non-centralized aid in the community, but both are mathematically equivalent in terms of redistributing wealth. As shown in Fig 2, the T model sets a new transfer rate $\xi$ and period $t_p$. Although the timing of inheritance and donation during life and after death differs from agent to agent, the T model assumes that $N$ agents simultaneously distribute the wealth $\xi \cdot m_i(t)$ corresponding to the transfer rate $\xi$ to all others equally in each period $t_p$. This is because in estimating redistribution's effect on reducing disparity, it is sufficient to establish an average time period and an average amount of redistribution. The wealth $m_i(t+\Delta)$ of agent $i$ at time $t+\Delta$ immediately after period $t_p$ is expressed as Eq (4):

$$m_i(t + \Delta) = (1 - \xi) \cdot m_i(t) + \xi \cdot \frac{\sum_{j \neq i} m_j(t)}{N - 1}. \qquad (4)$$

An equation similar to Eq (4) is found in a model that explains the firm size distribution [28]. In this model, a fraction of the firm's employees remains with a retention rate $\lambda$, and the sum of the $(1-\lambda)$ fraction of leavers (the leaver pool) is split with random probability $\varepsilon$ to the other firms. In the firm size model and Eq (4), the remainder and non-transfer terms and the leaver pool and transfer pool terms correspond to each other, but both models differ in that the leaver pool term is split with random probability $\varepsilon$, while the transfer pool term is equally distributed to all $(N-1)$ others. This is because the firm size model models random job switching among firms, whereas Eq (4) models redistribution to the community as a whole.

In addition, the concept of quantile is introduced for redistribution. I refer to this as the Q model for convenience. Quantiles are sometimes used to assess disparity [29] and are used here to estimate what level of wealth redistribution benefits the poor. In the Q model, Eq (5) is used to first calculate the number of agents $N_q$ whose wealth $m_i(t)$ is less than or equal to the 1/

T model (transfer)

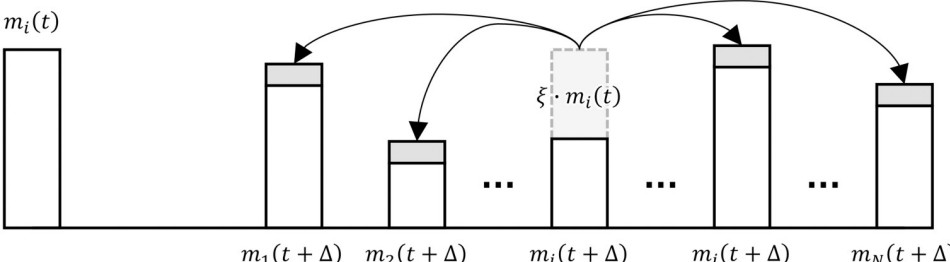

**Fig 2. Redistribution models.** T model (transfer), where $\xi$ is the transfer rate. The wealth $\xi \cdot m_i(t)$ is equally distributed from one agent to the other $N$ agents, and this distribution is carried out by all $N$ agents in every period $t_p$.

$q$ quantile of the wealth maximum $m_{MAX}(t)$ and then the total wealth $S_q$ of agents whose wealth $m_i(t)$ exceeds the $1/q$ quantile.

$$
\begin{aligned}
&m_{MAX}(t) = Max(m_i(t), i \in \{1, 2, \cdots, N\}), \\
&N_q = Count\left(m_i(t) \leq \frac{m_{MAX}(t)}{q}, i \in \{1, 2, \cdots, N\}\right), \\
&S_q = \sum_{\substack{m_i(t) > \frac{m_{MAX}(t)}{q}}} m_i(t).
\end{aligned}
\tag{5}
$$

Then, agent $i$ above the $1/q$ quantile transfers wealth $\xi \cdot m_i(t)$ corresponding to the transfer rate $\xi$, and agent $i$ below the $1/q$ quantile receives the redistributed wealth $\xi \cdot S_q/N_q$. The wealth $m_i(t+\Delta)$ of agent $i$ at time $t+\Delta$ immediately after period $t_p$ is expressed as Eq (6) according to wealth $m_i(t)$ at time $t$:

$$
\begin{aligned}
&if \ m_i(t) > \frac{m_{MAX}(t)}{q}, \\
&\quad m_i(t + \Delta) = (1 - \xi) \cdot m_i(t); \\
&if \ m_i(t) \leq \frac{m_{MAX}(t)}{q}, \\
&\quad m_i(t + \Delta) = m_i(t) + \xi \cdot \frac{S_q}{N_q}.
\end{aligned}
\tag{6}
$$

## Gini index

The Gini index is used to evaluate disparities due to exchange and redistribution. It is calculated by drawing a Lorenz curve and the equal distribution line [30]. Mathematically, the wealth $m_i(t)$ of $N$ agents at time $t$ is ordered from smallest to largest, and the Gini index $g$ is calculated using Eq (7). When the wealth of $N$ agents is perfectly equal (uniform distribution), the Gini index $g$ is 0. When all wealth is concentrated in a single agent (delta distribution), $g$ is 1. In other words, $g$ ranges from 0 to 1, and the larger the disparity, the larger the $g$ value.

$$
\begin{aligned}
&r_i(t) = Sort(m_i(t)), \\
&g = \frac{2 \cdot \sum_{i=1}^{N} i \cdot r_i(t)}{N \cdot \sum_{i=1}^{N} r_i(t)} - \frac{N + 1}{N}.
\end{aligned}
\tag{7}
$$

## Results

### Exchange

First, I examine the wealth distribution for the R model represented by Eq (1), the L model represented by Eq (2), and the J model represented by Eq (3). Fig 3 shows the simulation results. The results of the R model (Fig 3A) indicate that the wealth distribution approaches a gamma distribution as time $t$ elapses, which is consistent with previous studies [20]. The L model (Fig 3B) approaches a power-law distribution with extreme disparity as time $t$ elapses. The J model's (Fig 3C) profit/loss rate $\delta$ has the same width $\delta_w$ as the L model, but it has a looser exponential distribution than a power-law distribution, and the disparity is lower than that in the L model. Conversely, the J model in Fig 3D approaches a power-law distribution similar to the L model because the width $\delta_w$ of the profit/loss rate $\delta$ is larger than that in Fig 3C.

Next, I examine the Gini indices $g$ for the R, L, and J models using Eq (7). Fig 4 shows the simulation results. The R model shows that the Gini index $g$ generally converges to 0.4 as time

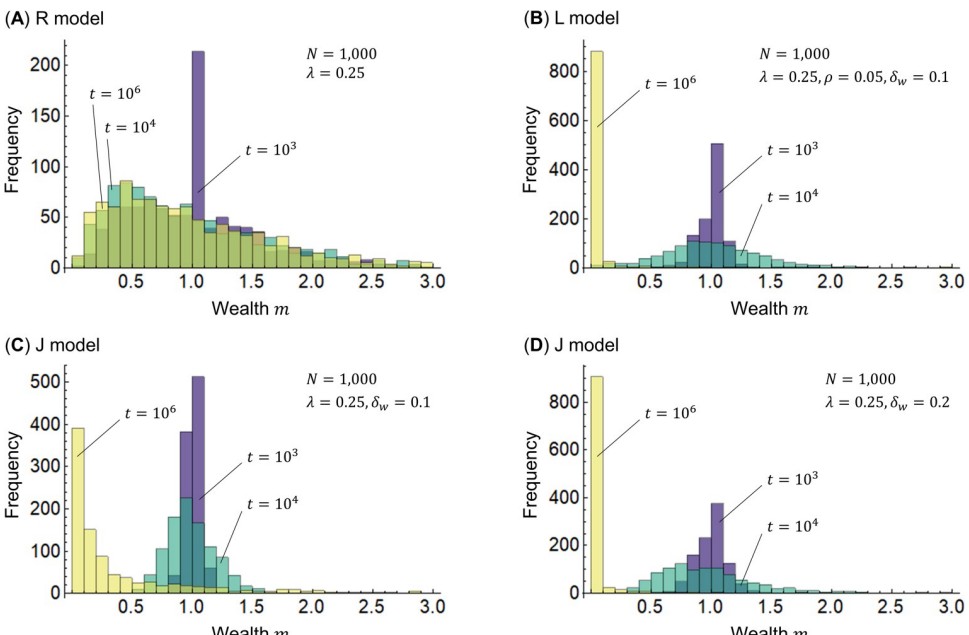

**Fig 3. Wealth distributions for the R, L, and J models.** (A) R model, (B) L model, and (C) and (D) J model. The horizontal axis represents wealth $m$, and the vertical axis represents frequency. In all models, the number of agents is $N = 1000$. The initial values of wealth at time $t = 0$ are $m_i(0) = 1$ ($i = 1,2,\cdots,N$). Since the average savings rate to GDP worldwide is roughly 0.25 [31], the savings rate here is $\lambda = 0.25$. In the L model, the interest rate is $\rho = 0.05$, and the width of the profit/loss rate $\delta$ is $\delta_w = 0.1$; in the J model, $\delta_w = 0.1$ and 0.2, so the effect of the profit/loss rate $\delta$ can be observed. To determine the change in wealth distribution, time (number of exchange repetitions) is $t = 10^3$, $10^4$, and $10^5$.

$t$ elapses, as expected based on previous research [20]. The L model has a Gini index $g$ approaching 1 regardless of the interest rate $\rho = 0$ or 0.05. In the J model, the Gini index $g$ approaches 1 slowly when the width of the profit/loss rate $\delta$ is $\delta_w = 0.1$, but when $\delta_w = 0.2$, the Gini index $g$ approaches 1 more quickly.

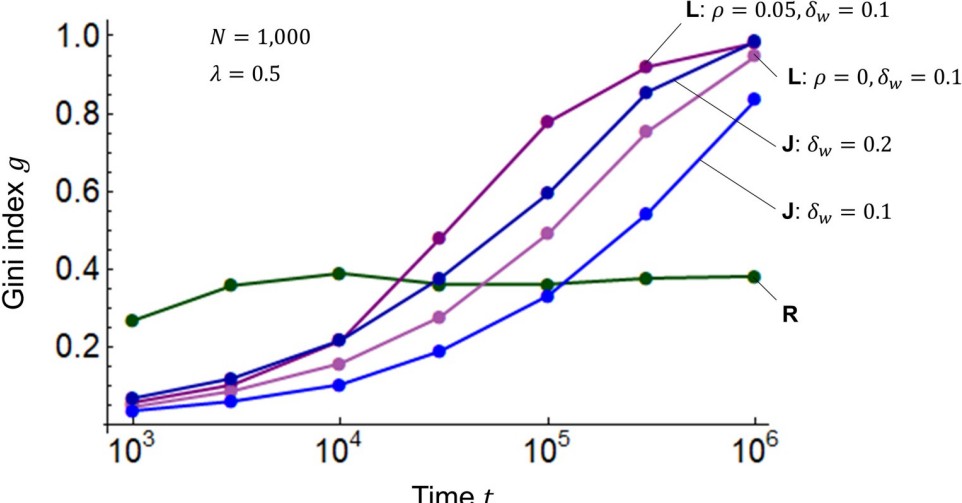

**Fig 4. Gini indices for the R, L, and J models.** The horizontal axis represents time $t$, and the vertical axis represents the Gini index $g$. In all models, the number of agents is $N = 1000$, the initial values of wealth are $m_i(0) = 1$ ($i = 1,2,\cdots,$ $N$), and the savings rate is $\lambda = 0.25$. The interest rates of the L model are $\rho = 0$ and 0.05, and the width of the profit/loss rate $\delta$ is $\delta_w = 0.1$. The widths of the J model are $\delta_w = 0.1$ and 0.2.

The Gini index $g$ converges to a small value in the R model because wealth is distributed between the poor and the wealthy with random division probability $\varepsilon$. Comparing the L and J models with the same width $\delta_w = 0.1$ and interest rate $\rho = 0$, we can see that the structure of the L model is the reason for the larger disparity; namely, only one of the agents bears the profit and loss in the L model. This suggests that joint ventures under shareholder capitalism and *mudaraba* are more effective in restraining disparity than financial capitalism, as well as the prohibition of *riba*. The fact that the Gini index $g$ approaches 1 either earlier or later in the L and J models suggests that exchange alone cannot prevent disparity from increasing and redistribution is essential. Further, the larger Gini index $g$ when $\delta_w = 0.2$ than when $\delta_w = 0.1$ (the width of the profit/loss ratio $\delta$ in the J model) suggests that speculative projects increase disparity, that is, the prohibition of *gharar* is effective; moreover, face-to-face *mudaraba* that promotes partnership is more effective at decreasing disparity than shareholder capitalism, as it would potentially restrain speculation.

## Redistribution

I combine the R model in Eq (1), the L model in Eq (2), and the J model in Eq (3) with the T model in Eq (4) to examine the Gini index $g$ using Eq (7) when wealth is redistributed via transfer. Fig 5 shows the simulation results. Using period $t_p$ for the transfer, I take $10^4$ when the Gini index $g$ begins to rise in Fig 4, and $10^5$ when $g$ exceeds the alert level of 0.4 for potential social unrest. The results of the R-T model are almost identical to those of the R model, regardless of redistribution or period $t_p$. This is because the R model itself distributes wealth between the poor and the wealthy. In the R-T, L-T, and J-T models, the Gini index $g$ converges from 0.4 to 0.9, which is less than 1, for the period $t_p = 10^5$, and from 0.1 to 0.3, which is even smaller than that for the period $t_p = 10^5$, for $t_p = 10^4$. The dependence of the Gini index $g$ on period $t_p$ is inferred to be approximately logarithmic. These results indicate that redistribution through transfer suppresses disparity and repeated transfers over a shorter period of time further suppress disparity.

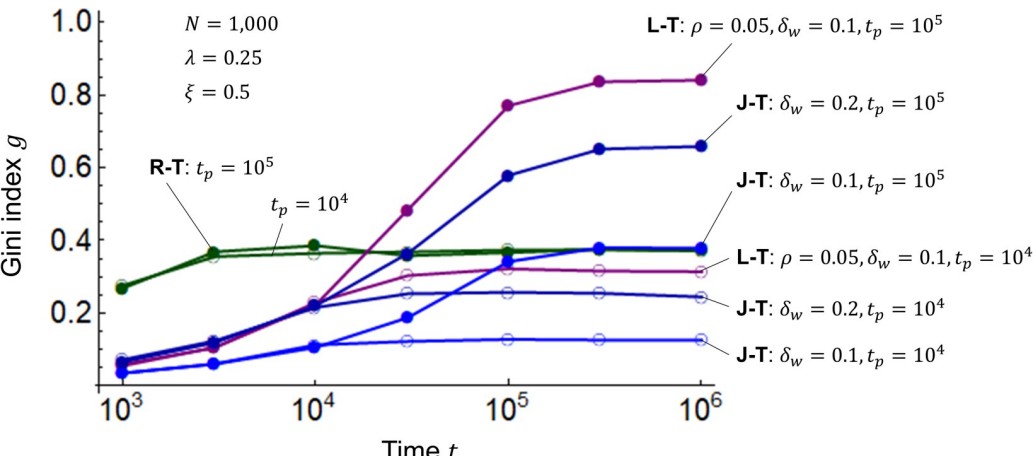

**Fig 5. Gini indices for the R-T, L-T, and J-T models.** The horizontal axis represents time $t$, and the vertical axis represents the Gini index $g$. The T model is combined with the R, L, and J models. In the R-T, L-T, and J-T models, the number of agents is $N = 1000$, the initial values of wealth are $m_i(0) = 1$ ($i = 1,2,\cdots,N$), and the savings rate is $\lambda = 0.25$. As the highest inheritance tax rate among OECD countries is roughly 0.5 [32,33], the transfer rate here is $\xi = 0.5$. The L-T model has an interest rate of $\rho = 0.05$ and a profit/loss rate $\delta$ with a width of $\delta_w = 0.1$. The widths of the J model are $\delta_w = 0.1$ and 0.2. The time periods for the transfers are $t_p = 10^4$ and $10^5$.

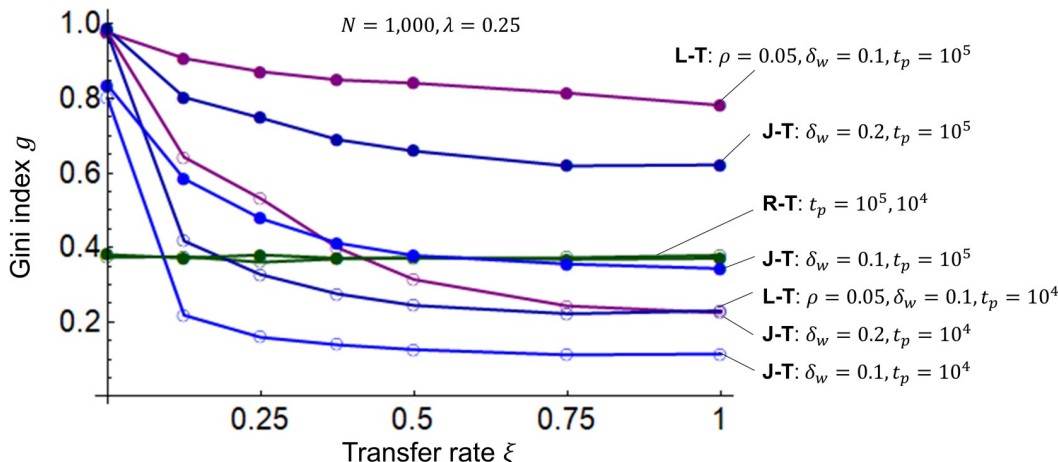

**Fig 6. Gini indices for the R-T, L-T, and J-T models.** The horizontal axis represents the transfer rate $\xi$, and the vertical axis represents the Gini index $g$. In the R-T, L-T, and J-T models, the number of agents is $N = 1000$, the initial values of wealth are $m_i(0) = 1$ ($i = 1,2,\cdots,N$), and the savings rate is $\lambda = 0.25$. The interest rate in the L-T model is $\rho = 0.05$, and the width of the profit/loss rate $\delta$ is $\delta_w = 0.1$ The widths for the J-T model are $\delta_w = 0.1$ and $0.2$. The time periods for the transfers are $t_p = 10^4$ and $10^5$.

I then investigate the dependence of the Gini index $g$ on the transfer rate $\xi$. Fig 6 shows the simulation results. In the R-T, L-T, and J-T models, the Gini index $g$ decreases rapidly until the transfer rate $\xi$ increases from 0 to approximately 0.2, but $g$ does not decrease if $\xi$ reaches approximately 0.5 or higher. In other words, it does not make sense to make $\xi$ unnecessarily large, since the effect of suppressing disparity is difficult to obtain when $\xi$ is greater than 0.5. Although the actual inheritance tax rate varies with the amount of inheritance and number of heirs, $\xi = 0.5$ is the lowest saturation point of $g$, which may correspond to the fact that the maximum tax rate in OECD countries is generally 0.5 [32,33]. Regarding *waqf*, I could not find any statistical data that correspond to an inheritance tax, but given the extremely significant role of *waqf* in public facilities and welfare in Islamic societies [34,35], and threfore I can infer that the value corresponding to the transfer rate $\xi$ is quite large. In addition, voluntary *waqf* in the Islamic economy is considered more meaningful for the well-being of both the individual and the community (*ummah*) than inheritance tax, which is mandated by the authorities in a capitalist society.

I combine the R, L, and J models with the Q model expressed in Eqs (5) and (6) to investigate the Gini index $g$ when wealth is redistributed according to quantile $1/q$. Fig 7 shows the simulation results. For 1/1 quantiles, the R-Q, L-Q, and J-Q models are equivalent to the R, L, and T models, respectively. The R-Q, L-Q, and J-Q models all generally approach the Gini index $g$ of the R-T, L-T, and J-T models between the 1/4 and 1/6 quantiles, respectively. This corresponds to the quintile axiom in welfare economics, which states that countries should focus on raising the welfare of the poorest one-fifth of the population [36,37]. The reason the Gini index $g$ of the J-Q model does not fall fully to the level of the J-T model when $\delta_w = 0.2$ is that the wealth distribution is unstable because of wealth fluctuations across quantiles, as $\delta_w$ is large. Further, the potential reason $g$ is larger inversely above the 1/8 quantile is that redistribution causes a reversal of wealth between the $1/q$ quantile, the poorest quantile, and the $2/q$ quantile, the next poorest quantile.

An overview of Figs 5–7 shows that exchange alone, as in the L and J models, widens disparity, and combining it with redistribution, as in the T and Q models, is essential. Using the Gini index $g = 0.4$ as a guide (i.e., warning level for social unrest) [2], I can show that 1) interest rate $\rho$ should be avoided, as in the finance capitalism L model; 2) speculation with a large profit/

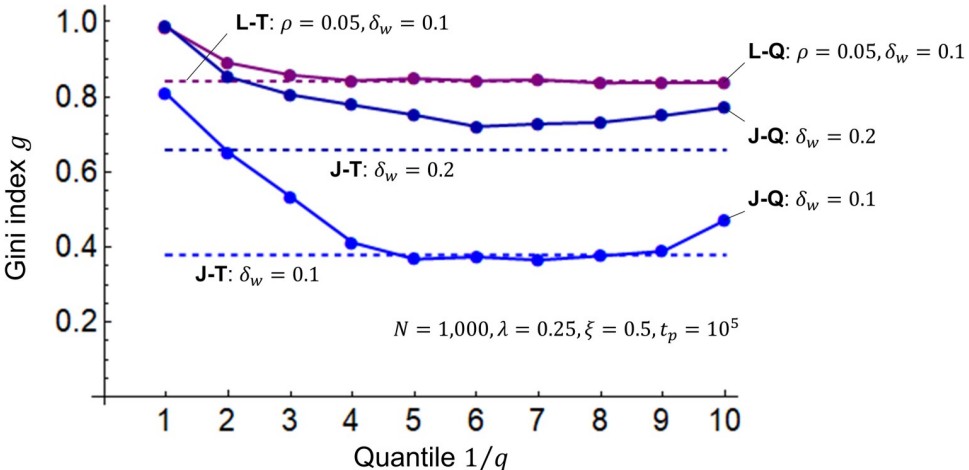

**Fig 7. Gini indices for the R-Q, L-Q, and J-Q models.** The horizontal axis represents the quantile $1/q$, and the vertical axis represents the Gini coefficient $g$. The Q model is combined with the R, L, and J models. In the R-Q, L-Q, and J-Q models, the number of agents is $N = 1000$, the initial values of wealth are $m_i(0) = 1$ ($i = 1,2,\cdots,N$), the savings rate is $\lambda = 0.25$, and the transfer rate is $\xi = 0.5$. The L-Q model's interest rate is $\rho = 0.05$, and the width of the profit/loss rate $\delta$ is $\delta_w = 0.1$. The widths of the J-Q model are $\delta_w = 0.1$ and $0.2$. The period over which the transfer takes place is $t_p = 10^5$.

loss width $\delta_w$ should be avoided, even in the joint venture J model; 3) the wealth in the inheritance tax and *waqf* T model should be redistributed with a transfer ratio $\xi$ in the range of 0.2 to 0.5; and 4) the interval $t_p$ of redistribution should be shortened from $10^5$ to $10^4$ as much as possible; further, according to the Q model, wealth should be redistributed to the poorest 1/4 to 1/6 quantiles of the population, as per the quintile axiom [36,37].

Note that in the R model expressed in Eq (1), the total amount of wealth is conserved even after repeated exchanges. In contrast, the L model in Eq (2) and the J model in Eq (3) do not conserve the total amount of wealth before and after exchange because of the random profit/loss rate $\delta$. In Figs 4–7, however, the simulation results do not change when the total wealth of the L and J models varied, because the relative relationships among agents are evaluated using the Gini index $g$. If one runs the simulation with a constant total amount of wealth, the amount of wealth for each agent at each exchange should be adjusted according to changes in the total amount of wealth. In addition, although I set a random number centered at 0 for the profit/loss ratio $\delta$, it is possible to reduce/increase the total amount of wealth by giving a negative bias in a recession and a positive bias in an economic recovery.

These findings are derived from econophysics models that, although simplifying the complex interaction phenomenon (e.g., human society and natural phenomena), show the inevitability of its occurrence under the laws of physics. It is impressive that although lacking an econophysics framework, Islamic society, based on trial and error over a millennium, was able to create a legal system based on the four main findings above. Incidentally, if time $t$ in these econophysics models corresponded to a millennium, it would be on the order of $10^{4\sim5}$ (12 months—365 days × 1000 years), and a human lifetime would correspond to $10^{3\sim4}$ (12 months—365 days × 10–100 years). The population of the medieval Islamic world is estimated to be on the order of $10^{6\sim7}$ [38], which, divided by the medieval population size of cities on the order of $10^{2\sim3}$ [39], is approximately on the order of $10^{3\sim4}$. Thus, combining time and population, the number of economic trials (i.e., the different economic systems attempted) would have been repeated on the order of $10^{4\sim5}$ times during the millennium. The Islamic world has experienced much social unrest throughout such trials, and it has developed a corresponding legal system to control disparity.

## Discussion

The results of the comparison of Islamic and capitalist economies using an econophysics approach provide quantitative support that prohibiting *riba*, which unevenly distributes wealth, promoting reciprocal joint ventures through *mudaraba*, and redistributing wealth through *waqf* help control disparity in Islamic economies. Using a long-term view of human history as reference, the modern era is in transition from a monetary economy to a credit economy; the financial innovations and ethical practices introduced by Islamic societies in the Middle Ages provide guidelines for how to establish a credit economy as the next alternative to capitalism.

These guidelines include a prohibition on financial transactions (interest and speculation), a return to an economy based on real transactions rooted in the production and exchange of goods and local communities, and the promotion of a face-to-face economy based on joint ventures and cooperatives. The guidelines also include the revival of a more ethical economy based on mutual aid that replaces taxes imposed by a centralized power and specific religions.

In his theory of gifts [40], Marcel Mauss explained that gift-giving should be based not on how much one gives and receives but on the assumption that people will help each other in times of need, and the giving across generations ensures the establishment of community. In his theory of mutual aid [41,42], Pyotr Kropotkin also showed that morality has its origins in social instincts and society progresses by combining mutual aid, justice, and morality. Further, he argued that in modern times, the state absorbed society, but in post-modern times, society needs to take back power from the state.

In his theory of debt [5], David Graeber presented three main moral principles of economic relationships (baseline communism, exchange, and hierarchy). Baseline communism is a relationship in which each person contributes according to his or her ability, and each person receives according to his or her needs. The process of exchange moves toward equivalence, often with an element of competition, by calculating profit and loss and with the awareness that the entire relationship can be dissolved. Hierarchical relationships are governed by a web of customs and precedents and do not tend to operate through reciprocity. Graeber thus argued that baseline communism will be the basis for what follows capitalism; market relations require the norms of community and mutual aid that typify the human economy; and debt resulting from exchange becomes problematic because its quantity is rigorously calculated, equivalence is demanded, and people are disconnected from their own social context.

In Nathalie Sarthou-Lajus' philosophy of debt [43], she positioned Mauss' gift, Kropotkin's mutual aid, and Graeber's communism as debts that do not need to be paid, that exist outside of equivalence and that are not debts, and as repayments to third parties that span generations. Further, both Graeber and Sarthou-Lajus stated that the distribution of wealth should be carried out as repayment to another in any manner they desire.

Reflecting on world history [44], Kojin Karatani presented four main modes of exchange as the various stages of the world system. Mode of exchange A constitutes reciprocity in civil society (gift–return), B is plunder and redistribution in an empire (submission–protection), C comprises commodity exchange in the capitalist economy (money–commodity), and D is the restoration of the reciprocal and mutual-aid relationships of mode of exchange A at a higher dimension. Mode of exchange D is of a higher dimension because it eliminates the negative aspect of communal constraints in mode A and retains the positive aspect of individual freedom and self-interest in mode C. In other words, in mode of exchange D, self-interest and altruism, both based on individual freedom according to Graeber and Sarthou-Lajus, are compatible.

Anarchism is an ideology in which individual freedom and communal solidarity are not contradictory, and a free and equal society is sought through mutual agreement. Graeber and Andrej Grubacic defined anarchism using four qualities: decentralization, voluntary association, mutual aid, and the network model [45]. The aforementioned ideas by Mauss, Kropotkin, Graeber, Sarthou-Lajus, and Karatani are consistent with anarchism in that they aim at a human-focused economy involving free exchange and redistribution while incorporating moral concepts of gift and mutual aid [46].

The Islamic economy's legal system encompasses politics, economics, and society; it successfully balances self-interest through its individual pursuit via profit–loss sharing instruments such as *mudaraba* (joint ventures) and *murabaha* (consensual contracts) while prohibiting *riba* (interest) and *gharar* (speculation), which cause disparity, and promoting altruism as mutual aid through *waqf*, *sadaqah* and *zakat* in the equal and noncentralized *ummah* (community) under God [6–8]. The Islamic economy provides meaningful guidelines for achieving anarchism, even though it is based on religion. In other words, the Islamic economy has the potential to transform capitalism into the next credit economy in modern times [25–27], just as it brought about moral and financial innovation in the Middle Ages. The challenge in the non-Islamic world, however, is not redistribution through taxes collected under a centralized power but redistribution through individuals' own free choice under a noncentralized community, as well as reconstruction of the social norm of mutual aid, that is, to make it possible for redistribution to occur without a specific religion.

In his account of the human history of violence and inequality, Walter Scheidel [47] argued that four types of events have reduced economic inequality: mass-mobilization warfare, transformative revolutions, state collapse, and catastrophic plagues. Currently, the world is suffering from the COVID-19 pandemic, war in Ukraine, and natural disasters and conflicts caused by global warming and its effects. While these are extremely unfortunate, they also help strengthen social connections and mutual aid in communities. Can we turn these crises into opportunities to rebuild morality-focused giving and mutual aid?

Environmental, social, and corporate governance investments [48,49] and social enterprises [50,51] are gaining popularity in economics. The former encourages repayment and mutual aid to third parties as mentioned, while the latter aims at an association economy linking communal reciprocity, public redistribution, and private market exchange. Meanwhile, the information industry is moving toward digital democracy [52,53] and platform corporativism [54,55]. The former aims for citizen participation in policy consensus and government services and the latter for joint ownership, fair profit sharing, and democratic governance. These movements may be a preliminary step toward moral restructuring.

## Conclusions

In this study, I proposed a new econophysics model of wealth exchange and redistribution to compare Islamic and capitalist economies and to determine guidelines for a credit economy, the next alternative to capitalism.

To model exchange, I took as parameters the savings rate $\lambda$, the interest rate $\rho$, and the profit/loss rate $\delta$ and developed a loan interest model to represent financial capitalism and *riba* and a joint venture model to represent shareholder capitalism and *mudaraba* of a *Shariah*-compliant profit–loss sharing system. In the loan interest model, one economic agent earns interest on the amount of wealth exchanged, excluding savings, while the other pays interest and bears all profits and losses on the amount exchanged by both combined. In the joint venture model, profits and losses are distributed in proportion to the amount exchanged by each economic agent.

To model redistribution, I took as parameters the transfer rate $\xi$ and period $t_p$ and created a transfer model that represents capitalism's inheritance tax and *waqf* of a *Shariah*-compliant wealth redistribution system. In this model, the amount of wealth corresponding to the transfer rate $\xi$ is equally distributed to all others in each period $t_p$.

Based on simulations using the exchange model, I found that prohibiting interest in the Islamic economy is effective in reducing disparity and the structure in which only one economic agent bears the profit and loss increases disparity in the loan interest model. Based on simulations using the joint venture model, exchange alone, even a reciprocal joint venture, cannot avoid increasing disparity, and redistribution is essential. As for the difference in the profit/loss ratio, the results showed that prohibiting *gharar* and promoting parties' active involvement in each other's affairs in *mudaraba* deter the speculation that would normally increase disparity.

Based on simulations combining the exchange and transfer models, the results showed that the more the transfer rate is increased from 0.2 to 0.5 and the period is shortened in wealth redistribution, the higher the disparity-control effect. As for the quantile model, the results supported the quintile axiom in welfare economics.

Note that although shareholder capitalism and *mudaraba* are mathematically represented by the same model, the latter is qualitatively more desirable because there is more active, face-to-face involvement than in the former. Additionally, although the inheritance tax and *waqf* are mathematically identical, the latter is again qualitatively more desirable because the former is mandatory and enforced through centralized power, while the latter is voluntary and based on the noncentralized aid of the *ummah*.

This study quantitatively supports the Islamic economy's superiority over capitalism. The next economic system to follow the current capitalist system should return to an economy based on real transactions, promote a face-to-face association economy through joint ventures and cooperatives, and revive an economy rooted in mutual aid as a response to power exclusive to the state and specific religions. These insights continue the lineage of Mauss' gift, Kropotkin's mutual aid, Graeber's baseline communism, Sarthou-Lajus' repayment to third parties, and Karatani's mode of exchange D, which all lead toward the ideal of anarchism.

The present study modeled only basic exchange and redistribution processes to clarify the fundamental differences between the Islamic economy and capitalism; I recommend a detailed analytical study in the future that incorporates different parameters: for example, setting the savings rate $\lambda$, interest rate $\rho$, and profit/loss rate $\delta$ according to the actual conditions of states and communities and setting the transfer rate $\xi$ and the period $t_p$ according to the various asset management and social security systems.

Note that a comparison of wealth/income Gini index findings between Islamic and non-Islamic countries (e.g., [56,57]) shows no striking differences between the two at first glance. However, it is possible that public goods by *waqf*, *sadaqah* or *zakat* are not reflected in the findings of the Gini index in Islamic countries, and this issue should be explored in future research. Moreover, while the present study utilizes an econophysics approach to indicate some of the underlying causes of disparity in exchange and redistribution, it is limited in its recommendations of practical policy or effective policy implementation. Such attempts remain for future empirical studies in economics, political science, and sociology.

## Acknowledgments

The author received valuable advice from the Hitachi Kyoto University Laboratory of the Kyoto University Open Innovation Institute regarding how to pursue this research. The author would like to express their deepest gratitude. The author also thanks very much the academic

editor and reviewers for their helful suggestions on the Islamic economy and the kinetic exchange models, and Dr. Pascal and Editage (www.editage.com) for English language editing.

## Author Contributions

**Conceptualization:** Takeshi Kato.

**Data curation:** Takeshi Kato.

**Formal analysis:** Takeshi Kato.

**Investigation:** Takeshi Kato.

**Methodology:** Takeshi Kato.

**Software:** Takeshi Kato.

**Validation:** Takeshi Kato.

**Visualization:** Takeshi Kato.

**Writing – original draft:** Takeshi Kato.

**Writing – review & editing:** Takeshi Kato.

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
