## [Decision Letter · Decision Letter 0]

24 Aug 2022

PONE-D-22-18743Islamic and capitalist economies: Comparison using econophysics models of wealth exchange and redistributionPLOS ONE

Dear Dr. Kato,

Thank you for submitting your manuscript to PLOS ONE. After careful consideration, we feel that it has merit but does not fully meet PLOS ONE’s publication criteria as it currently stands. Therefore, we invite you to submit a revised version of the manuscript that addresses the points raised during the review process.

As you will see in this mail, two experts provided positive reports for your work with whom I agree. In particular, please clarify how your equation (4) is related to Chakrabarti (2012, Physica A).

I am happy to accept your manuscript for publication in PLOS one after a minor revision to fully address the comments from two reviewers. I have one more point. The jounral titles in the reference list are not consistently abbreviated. Full titles are better.==============================

We look forward to receiving your revised manuscript.

Kind regards,

Dao-Zhi Zeng

Academic Editor

PLOS ONE

Journal Requirements:

Reviewers' comments:

Reviewer's Responses to Questions

**Comments to the Author**

1. Is the manuscript technically sound, and do the data support the conclusions?

Reviewer #1: Yes

Reviewer #2: Yes

2. Has the statistical analysis been performed appropriately and rigorously? 

Reviewer #1: N/A

Reviewer #2: Yes

3. Have the authors made all data underlying the findings in their manuscript fully available?

Reviewer #1: Yes

Reviewer #2: Yes

4. Is the manuscript presented in an intelligible fashion and written in standard English?

Reviewer #1: Yes

Reviewer #2: Yes

5. Review Comments to the Author

Reviewer #1: This study is a thoughtful and fascinating examination of Islamic and capitalist economies. In order to compare these economies and to lay forth principles for a credit economy—the next alternative to capitalism—the author of this paper suggests a new econophysics model of wealth exchange and redistribution. In light of wealth distribution and disparity reduction, the article is both interesting and significant. The author empirically demonstrates the ability of Islamic economics to redistribute wealth more equally. The literature review is sufficiently comprehensive, correlating with the author’s thesis. The methodology employed is novel in this field and, despite some criticism from other scientists, I think it gives sufficient instrumentation in this research. The data are well presented and discussed. The conclusions are clear and coherent. Although the overall impression of the article is excellent, here are a few comments that I believe will strengthen the article.

Abstract and Introduction

The abstract begins with the fundamental differences between capitalist and Islamic economies. Although true, the author's statement in this regard could be refined.

1. Line 21-23: The absence of riba is indeed a fundamental fact distinguishing Islamic from other economies. Mudaraba, on other hand, is just one of the numerous Shariah-compliant contracts (musharaka, murabaha, salam, etc.) which work on the profit-loss-sharing principle. The profit-loss-sharing principle is the other fundamental difference between Islamic and capitalist economy. Mudaraba, for itself, is not a fundamental distinction; it is just part of the whole, a segment, an instrument. Is the author specifying only mudaraba or can he generalize his conclusion to all contracts (moreover, later in the text he mentioned murabaha)? If the outcomes are only on a mudaraba basis, this should be indicated expressly. Zakat (also spelled zakah) is one of the five pillars of Islam. It is fundamental to the Islamic economy (along with riba and the profit-loss-sharing trade system based on real assets). Together with waqf and sadaqah (or shadaqah) they form a virtuous wealth distribution. Zakat is obligatory and the others are not.

Discussion

2. Line 473: Proceeding from my conclusion made for Line 21-23, the question arises here about mudaraba and murabaha. Are these two Islamic financial instruments the only two with which the Islamic economy "successfully balances self-interest through its individual pursuit"? Perhaps, it would be more appropriate to use the term “profit-loss-sharing instruments”.

Minor issues

3. Line 143: “… showed that that a delta…” (unnecessary repetition). To my knowledge, this phrase from the article has one more "that" than needed. If the author thinks so, the second should be dropped out.

Reviewer #2: The manuscript contains an analysis of two types of economic systems - Islamic and capitalist, viewed through the lens of kinetic exchange models (KEM henceforth) of markets. The authors show that the repeated exchange mechanism from the KEM is useful to delineate emergence of different distributional features and they shed light

on how inequality evolves. I found the idea interesting and worth pursuing.

I have attached a more detailed review report.

6. PLOS authors have the option to publish the peer review history of their article (what does this mean?). If published, this will include your full peer review and any attached files.

Reviewer #1: No

Reviewer #2: No

---

## [Author Response · Author response to Decision Letter 0]

1 Sep 2022

September 2, 2022

Dao-Zhi Zeng

Academic Editor

PLOS ONE

Dear Editor and Reviewers:

Thank you for your suggestions and comments on my manuscript titled “Islamic and capitalist economies: Comparison using econophysics models of wealth exchange and redistribution.” I appreciate the time and effort you and each of the reviewers have dedicated to providing insightful feedback on ways to strengthen the paper. I have incorporated changes that reflect the detailed suggestions you have so graciously provided. I hope that my edits and the responses provided below satisfactorily address all the issues and concerns you and the reviewers have noted.

To facilitate your review of my revisions, the following is a point-by-point response to the questions and comments delivered in your letter [PONE-D-22-18743].

Response to Editor:

1. Please clarify how your equation (4) is related to Chakrabarti (2012, Physica A).

Response: Thank you for your helpful suggestion. I have added Chakarabarti (2012, Physica A) to the references and added the difference between equation (4) and the firm size model in the text (p. 12, lines 254–262, reference: p. 30, lines 697–699).

2. The jounral titles in the reference list are not consistently abbreviated. Full titles are better.

Response: Thank you for your helpful suggestion. I have corrected all journal titles in the reference list to full titles (p. 27–33, References section).

Response to Reviewer #1:

1. Line 21-23: The absence of riba is indeed a fundamental fact distinguishing Islamic from other economies. Mudaraba, on other hand, is just one of the numerous Shariah-compliant contracts (musharaka, murabaha, salam, etc.) which work on the profit-loss-sharing principle. The profit-loss-sharing principle is the other fundamental difference between Islamic and capitalist economy. Mudaraba, for itself, is not a fundamental distinction; it is just part of the whole, a segment, an instrument. Is the author specifying only mudaraba or can he generalize his conclusion to all contracts (moreover, later in the text he mentioned murabaha)? If the outcomes are only on a mudaraba basis, this should be indicated expressly. Zakat (also spelled zakah) is one of the five pillars of Islam. It is fundamental to the Islamic economy (along with riba and the profit-loss-sharing trade system based on real assets). Together with waqf and sadaqah (or shadaqah) they form a virtuous wealth distribution. Zakat is obligatory and the others are not.

Response: Thank you for your helpful suggestion. I have added Shariah-compliant profit–loss sharing (mudaraba, murabaha, salam, etc.) and wealth redistribution (waqf, sadaqah and zakat), and added that I will take up mudaraba and waqf among them (p. 2, lines 22–25, lines 29-31, p. 6, lines 114–119, p. 24, line 554, p.25, lines 564-565).

2. Line 473: Proceeding from my conclusion made for Line 21-23, the question arises here about mudaraba and murabaha. Are these two Islamic financial instruments the only two with which the Islamic economy "successfully balances self-interest through its individual pursuit"? Perhaps, it would be more appropriate to use the term “profit-loss-sharing instruments.”

Response: Thank you for this suggestion. I have followed your advice and revised the text to “profit–loss sharing instruments such as mudaraba (joint ventures) and murabaha (consensual contracts)” (p. 23, lines 512–513). 

3. Line 143: “… showed that that a delta…” (unnecessary repetition). To my knowledge, this phrase from the article has one more "that" than needed. If the author thinks so, the second should be dropped out.

Response: Thank you for this suggestion. I have dropped out the second “that” (p. 7, line 147).

Response to Reviewer #2:

1. It is known point of debate in the context of KEM that generally all of such models are conservative in nature - in the sense that, the total amount of money ∑_i▒〖m_i (t) 〗 remains constant for all t. The current approach I think also sheds light on how one can think about non-conservative models. I advise the author to explore this point further.

Response: Thank you for this suggestion. I have added that the L and J models are non-conservative with respect to the total amount of wealth, and that even if they are non-conservative, there is no problem with the simulation results of the Gini index (p. 19, lines 420–425).

2. Relatedly, it is not entirely clear to me how the author is ensuring that the system does not explode when the model is non-conservative (each round of trading may add up to more or less wealth than before the trading occurs).

Response: Thank you for this suggestion. I have added a brief note on how to simulate with a constant total amount of wealth, and added the simulation of recession and recovery (pp. 19–20, lines 425–431).

3. The model given by equation 4 has been explored in Effects of the turnover rate on the size distribution of firms: An application of the kinetic exchange models (Chakrabarti, Physica A, 391(23) 1, 2012, 6039-50) although in the context of applying KEM in explaining firm size distributions. A parallel can be drawn here.

Response: Thank you for this suggestion. As noted in the response to the editor above, I have added the difference between equation (4) and the firm size model (p. 12, lines 254–262).

4. I am curious about the distinction the author has made between Islamist economies and capitalist economies. While the nature of transaction may differ, does the eventual distribution of wealth/income also differ across them? If you have some data on income/wealth in some Islamic country and some capitalist country, it would be useful to compare their empirical distributions to motivate the idea. I suspect that the distributions will not differ systematically across countries. If

so, the author can explore why the distributions do not differ although the nature of transactions differ.

Response: Thank you for this suggestion. I have added examples of the Gini index findings in Islamic and non-Islamic countries, and added a future issue to explore why there is no striking differences between the two (p. 26, lines 601–605).

Thank you, once again, for giving me the opportunity to strengthen the manuscript with your valuable comments and queries. I have worked hard to incorporate your feedback and hope that these revisions persuade you to accept my submission.

Sincerely,

Takeshi Kato

Hitachi Kyoto University Laboratory, Open Innovation Institute, Kyoto University

Yoshidahonmachi, Sakyo-ku, Kyoto-shi, Kyoto 606-8501, Japan

+81-75-753-9716

kato.takeshi.3u@kyoto-u.ac.jp

---

## [Decision Letter · Decision Letter 1]

12 Sep 2022

Islamic and capitalist economies: Comparison using econophysics models of wealth exchange and redistribution

PONE-D-22-18743R1

Dear Dr. Kato,

We’re pleased to inform you that your manuscript has been judged scientifically suitable for publication and will be formally accepted for publication once it meets all outstanding technical requirements.

Kind regards,

Dao-Zhi Zeng

Academic Editor

PLOS ONE

Additional Editor Comments (optional):

Two referees have confirmed that all points raised by them are well addressed in the revision. Accordingly, I am happy to inform you that the manuscript is accepted for the publication. Please  address the following two minor issues when you prepare the final version.

1. In your acknowledgements part, please include acknowledging comments received from the reviewers. Their volunteer work are very important to maintain the quality of your research.

2. In line 257 of p.12, delete "the" before "both models."

I look forward to the publication and citations of your paper in future.

Reviewers' comments:

Reviewer's Responses to Questions

**Comments to the Author**

1. If the authors have adequately addressed your comments raised in a previous round of review and you feel that this manuscript is now acceptable for publication, you may indicate that here to bypass the “Comments to the Author” section, enter your conflict of interest statement in the “Confidential to Editor” section, and submit your "Accept" recommendation.

Reviewer #1: All comments have been addressed

Reviewer #2: All comments have been addressed

2. Is the manuscript technically sound, and do the data support the conclusions?

Reviewer #1: Yes

Reviewer #2: (No Response)

3. Has the statistical analysis been performed appropriately and rigorously? 

Reviewer #1: Yes

Reviewer #2: (No Response)

4. Have the authors made all data underlying the findings in their manuscript fully available?

Reviewer #1: Yes

Reviewer #2: (No Response)

5. Is the manuscript presented in an intelligible fashion and written in standard English?

Reviewer #1: Yes

Reviewer #2: (No Response)

6. Review Comments to the Author

Reviewer #1: I read the revised version of the manuscript with great interest. All of my suggestions have been considered. The author's work has completely satisfied and impressed me. Overall, I believe the revised version has improved significantly.

Reviewer #2: (No Response)

7. PLOS authors have the option to publish the peer review history of their article (what does this mean?). If published, this will include your full peer review and any attached files.

Reviewer #1: No

Reviewer #2: No

---

## [Editor Report · Acceptance letter]

14 Sep 2022

PONE-D-22-18743R1 

Islamic and capitalist economies: Comparison using econophysics models of wealth exchange and redistribution 

Dear Dr. Kato:

I'm pleased to inform you that your manuscript has been deemed suitable for publication in PLOS ONE. Congratulations! Your manuscript is now with our production department. 

Kind regards, 

on behalf of

Dr. Dao-Zhi Zeng 

Academic Editor

PLOS ONE